# Multi-view clustering by CPS-merge analysis with application to multimodal single-cell data

**Lixiang Zhang**[1], **Lin Lin**[2], **Jia Li**[1] *

**1** Department of Statistics, The Pennsylvania State University, University Park, Pennsylvania, United States of America, **2** Department of Biostatistics and Bioinformatics, Duke University, Durham, North Carolina, United States of America

* jiali@psu.edu

## Abstract

Multi-view data can be generated from diverse sources, by different technologies, and in multiple modalities. In various fields, integrating information from multi-view data has pushed the frontier of discovery. In this paper, we develop a new approach for multi-view clustering, which overcomes the limitations of existing methods such as the need of pooling data across views, restrictions on the clustering algorithms allowed within each view, and the disregard for complementary information between views. Our new method, called *CPS-merge analysis*, merges clusters formed by the Cartesian product of single-view cluster labels, guided by the principle of maximizing clustering stability as evaluated by CPS analysis. In addition, we introduce measures to quantify the contribution of each view to the formation of any cluster. CPS-merge analysis can be easily incorporated into an existing clustering pipeline because it only requires single-view cluster labels instead of the original data. We can thus readily apply advanced single-view clustering algorithms. Importantly, our approach accounts for both consensus and complementary effects between different views, whereas existing ensemble methods focus on finding a consensus for multiple clustering results, implying that results from different views are variations of one clustering structure. Through experiments on single-cell datasets, we demonstrate that our approach frequently outperforms other state-of-the-art methods.

## Author summary

Advances in single-cell profiling technologies have made it possible to measure various types of features from a single cell. In this new type of data, known as multimodal single-cell data, each cell has numerical measurements from multiple views. Analyzing multimodal data has opened up new horizons for single-cell genomics, where clustering is a fundamental analysis for validating existing hypotheses or discovering insights when little prior knowledge is available. Existing clustering methods either combine data from different modalities for simultaneous processing or use integration algorithms to aggregate clustering results from multiple views. In this paper, we propose a new approach called CPS-merge analysis, which considers both consensus and complementary effects among clustering results across views and provides a quantified contribution of each view. The

**Data Availability Statement:** All relevant datasets are described within the manuscript and Supporting information. The code is available at https://github.com/LixiangZhang/CPS-merge.

**Funding:** The research is supported by the National Science Foundation (NSF) under grant DMS-2013905. LZ and JL received summer salary from this NSF grant. The funder had no role in study design, data collection and analysis, decision to publish, or preparation of the manuscript.

**Competing interests:** The authors have declared that no competing interests exist.

approach operates on single-view cluster labels, enabling the use of advanced clustering algorithms in any individual view. Furthermore, since CPS-merge analysis does not require pooling the original data, it can be applied to distributed sources or data with sharing concerns. This new approach tackles the problem of multi-view clustering from a novel combinatorial perspective and has the potential to become a widely used and effective tool.

## Introduction

Multi-view data are becoming increasingly prevalent in real-world applications. For example, a data entry of a subject can contain image, audio, and text data. Single-cell genomics is a prominent biomedical area where multi-view data often arise. In the literature on single-cell data analysis, the term "multimodal" is usually used instead of "multi-view". In this paper, we use them interchangeably. To apply our methods developed here, data in different views can be of different modalities. For instance, RNA expression levels of cells can reveal much of the cellular heterogeneity, and many advanced techniques and tools have been developed to analyze such data, *e.g.*, Matrix factorization [1] for revealing low-dimensional structure, CIDER [2] for clustering. However, other data modalities, *e.g.*, DNA, protein expression, are found necessary to fully understand the cellular mechanics [3, 4]. RNA expression is often inadequate to separate immune cells that are molecularly similar but functionally distinct, and many subpopulations of T cells, indistinguishable by scRNA-seq data, are identifiable in other modalities [5, 6]. These examples present the following case: each view contains useful information about an instance, and the information in different views is complementary to some degree. A well-designed learning algorithm that leverages all the views can greatly improve performance. In particular, analytical tools for multimodal single-cell data have helped reconstruct gene regulatory networks, a significant leap forward for revealing the inner workings of biological systems [7, 8]. In biomedical multi-view learning, several related but different tasks have been pursued, *e.g.*, multi-view classification [9], multi-view clustering [10], multi-view deconvolution [11], and multi-view data integration [12]. Here, we focus on unsupervised multi-view clustering, used to reveal the underlying cellular structure that can assist downstream analysis.

The authors of [13] proposed to divide multi-view clustering methods for genomics data into three types: early, intermediate, and late integration. The early integration type contains methods that concatenate variables across all the views. Many drawbacks, *e.g.*, the sharp increase in dimension, the neglect of special statistical properties of particular views, have been noted for such methods [14]. These issues, especially severe for multimodal single-cell data, are mitigated to some extent by intermediate integration methods. According to a few highly regarded surveys on multi-view learning [15–18], intermediate integration methods include well-known multi-view clustering algorithms that belong to four schools: co-training, multiple kernel clustering, multi-view subspace clustering, and multi-view graph clustering. These methods combine data from multiple views into one set using weights, transformations, or simplification based on similarity or dimension reduction. In contrast, the late integration methods, which mostly belong to ensemble clustering methods, generate aggregated clusters based on clustering results obtained in every single view. Example methods of late integration ensemble clustering include [19–22]. Specifically, in [21, 22], dissimilarity measures between clusters in different results are computed based on the cluster membership of samples in each result. Dendrogram clustering is then applied to yield an integrated clustering result.

Ensemble clustering methods are popular for treating multi-view data, for example, the fast multi-view clustering via ensembles (FastMICE) method [23]. However, not all ensemble methods strictly adhere to the early, intermediate, and late integration taxonomy. FastMICE [23] is of particular interest because it employs a hybrid of early and late integration. This method of mixed early-late integration aims to identify a consensus among view-group clustering results, with each view group containing a random number of views, such as a single view or multiple views. The clusters in multi-view groups may be established using early integration.

Late integration methods overcome some disadvantages of early or intermediate integration [19–22]. Apparently, it is straightforward for such methods to incorporate advanced clustering methods developed for single-view data since they operate on single-view cluster labels instead of the original data. Consequently, late integration methods are easier to adopt when data in multiple views cannot be pooled, for instance, due to privacy concerns. On the other hand, the advantages of late integration come at a cost. Since such methods only examine the single-view cluster labels for integration, naturally, relevant information in the original data but not retained in cluster labels cannot be leveraged.

There are two primary principles for multi-view clustering, namely, the *consensus principle* [24] and the *complementary principle* [15]. The consensus principle assumes a shared clustering structure across all views, so clustering results from different views are considered variations of a single clustering result. Methods developed under this principle seek a plausible "average" of the clustering results. In contrast, the complementary principle emphasizes that clusters may only emerge when data from all views are analyzed together, as is often observed with single-cell data. Early integration methods naturally follow this principle since original data from all views are combined. However, incorporating the complementary principle into late integration methods is challenging because they only have access to cluster labels from different views. In fact, existing late integration clustering methods ignore the complementary effect between views.

Most methods in the intermediate integration type also ignore the complementary principle. For instance, multi-view clustering algorithms by co-training [25, 26] make the underlying assumptions of sufficiency and compatibility: (a) each view is sufficient for clustering on its own, (b) the target function of both views predict the same labels for co-occurring features with a high probability, and so on. Under the sufficiency assumption, co-training methods aim at maximizing agreement between two views (consensus principle). In addition, the compatibility assumption restricts clustering algorithms allowed. Specifically, similar algorithms are used in different views. Not only co-training methods but also other methods in the intermediate integration type, *e.g.*, multiple kernel clustering [27–29], multi-view subspace clustering [30–33], and multi-view graph clustering [34–36], are by construction not ready to leverage state-of-the-art algorithms for clustering single-view data. Furthermore, the multiple kernel clustering methods do not scale well with the sample size due to the quadratic complexity (in sample size) for computing the kernel matrix. The multiple subspace clustering methods assume implicitly that a shared latent subspace across the views determines the clusters (the spirit of the consensus principle). Multi-view graph clustering methods, aiming at finding a fusion graph from multiple views, are vulnerable to noisy datasets because it ignores inconsistency between views [37]. A number of algorithms have been designed specifically for multi-modal single-cell data, *e.g.*, weighted-nearest neighbor (WNN) analysis [5], totalVI [38], and multi-omics factor analysis v2 (MOFA+) [39]. Based on the comparison in [5], WNN is the state-of-the-art multimodal single-cell clustering algorithm, but it is a multi-view graph clustering approach with disadvantages discussed previously. Last but not least, the intermediate

integration methods must centralize the multiple views at the data level and hence are not applicable to distributed data.

In this paper, we aim at developing a late integration method that accounts for both the consensus and complementary principles. Although late integration has many benefits for single-cell data, existing methods overlook the importance of the complementary principle. We illustrate the significance of this principle through three simulated scenarios, with detailed findings provided in Fig A, Fig B, and Fig C in S1 Appendix. Our results demonstrate that the complementary effect between views plays an essential role for identifying meaningful clusters.

Our novel algorithm called *Covering Point Set Merge* (*CPS-merge*) analysis contributes to the paradigm of late integration by combining the two principles, namely, the consensus and complementary principles. In CPS-merge analysis, we create Cartesian product clusters based on single-view clusters. Many product clusters may arise due to randomness and may not represent meaningful subgroups. To address this issue, we have developed a computationally efficient approach to merge product clusters by considering the uncertainty level of each cluster. Furthermore, we propose a new measure to quantify the contribution of each view to the identification of any final cluster. This measure is valuable for understanding cell heterogeneity in single-cell studies.

## Materials and methods

### Overview of analysis pipeline

The pipeline of CPS-merge analysis is shown in Fig 1. CPS-merge analysis generates an aggregated multi-view clustering result by the following modules.

- *Module 1*: Data are perturbed by random noises in each view. A collection of clustering results (aka, partitions) are obtained from the perturbed data using a view-specific and user-specified clustering algorithm. The same algorithm is applied to the original data to yield a clustering result which we call *reference partition*. Then clusters in different clustering results are aligned with the reference partition via optimal transport, a step to remove inconsistency in the cluster labels used in different results.

- *Module 2*: We form new clusters by the Cartesian product of the clusters from two or more views, that is, each ordered pair (or *k*-tuples in general) of cluster labels from the two views defines one cluster.

- *Module 3*: To obtain a final clustering result, we merge unstable clusters progressively to maximize *tightness* given a specified number of final clusters. The tightness measure is defined in [40], which quantifies the clustering stability. A comprehensive review of clustering stability is referred to [41]. If the number of Cartesian product clusters at the start of merging is large (for example, more than 100), we conduct a *first-stage merging* by bipartite clustering. Otherwise, we directly begin the *second-stage merging* using Covering Point Set (CPS) analysis, available via the R package `OTclust` [40].

The output of CPS-merge analysis contains an integrated clustering result and quantities that measure the contribution of each view to the final clusters. The Cartesian product clusters from multiple views capture the complementary effects between the views. On the other hand, these clusters are subject to merging based on cross-view correspondence between clusters. We establish this correspondence by CPS analysis under the consensus principle. The cross-view correspondence exists as a mapping between clusters or between the so-called super-clusters, the former by optimal transport and the latter by bipartite clustering.

**Fig 1. The pipeline of CPS-merge analysis.** When there are more than two views, users can either directly treat the Cartesian product clusters with higher orders or conduct step-wise merging such that two views are treated at each step. Current mutlimodal single-cell datasets only contain two views.

The computation complexity of CPS-merge analysis depends on all the modules. In Module 1, the complexity of generating perturbed data is linear in sample size. The complexity of the single-view clustering algorithms used can vary. However, many clustering algorithms have linear complexity in sample size. In Module 2 and 3, CPS-merge analysis involves optimal transport or bipartite clustering applied to the single-view clusters instead of the original data points. Hence the complexity is quadratic in the number of clusters, which is usually much smaller than the sample size. In summary, if the single-view clustering algorithms have linear complexity in sample size, the complexity of CPS-merge analysis will also be linear unless the number of clusters is in the same order as the sample size.

Although there are usually two views in current multimodal single-cell datasets, our method extends straightforwardly to more than two views. In such a case, our method can be applied either directly to the Cartesian product clusters across all the views or progressively to two views at a time, *e.g.*, aggregating the first two views into one and then taking the third view as the other view, so on so forth. Without loss of generality, we assume there are two views in our discussion. Next, we elaborate on each module in CPS-merge analysis.

## Module 1: Generate coherent cluster labels within each view

Because our algorithm aims at maximizing the overall tightness of a clustering result, we need to generate random variations of a clustering result within each view to evaluate tightness. The tightness measure is defined for both individual clusters and the entire partition to quantify the level of uncertainty. We first explain in the steps listed below how to obtain random variations of a clustering result. Details for the definition and computation of the tightness of a cluster are provided at the end of this Section.

1. Apply clustering to the original single-view data and call the result *reference partition*.

2. Perturb the original single-view data by adding random noise to each point. The noise is sampled from a Gaussian distribution with mean zero and variances adjusted with the data. We usually set the variance to be 10% of the average within-cluster variance. Repeat the perturbation step to obtain multiple perturbed versions of the whole dataset.

3. Obtain a collection of partitions by applying a user-chosen clustering algorithm to every perturbed dataset.

4. Align clusters in those partitions with the clusters in the reference partition.

Note that our algorithm works with any baseline clustering algorithm chosen for a single view. Thus users can easily incorporate state-of-the-art clustering algorithms. Because of the unsupervised nature of clustering, Step (4) to align clusters (the reference partition as a benchmark) is necessary since cluster labels used in different partitions are not automatically

consistent. For instance, even if two partitions are identical, the cluster labels may be permuted. In practice, precise correspondence between clusters in two partitions rarely exists. We use a cluster alignment algorithm based on optimal transport [42]. Next, we explain how clustering results are aligned within each view. More details can be found in [42].

Suppose there are two partitions denoted by $\mathcal{P}^{(p)}$, $p = 1, 2$, each contains $k_p$ clusters $C_1^{(p)}, \ldots, C_{k_p}^{(p)}$. The alignment between clusters is captured by a so-called *cluster aligning matrix*:

$$W = (w_{i,j})_{i=1,\ldots,k_1; j=1,\ldots,k_2}, \quad w_{i,j} \in [0,1].$$

The entry $w_{i,j}$ is a coupling/matching weight between $C_i^{(1)}$ and $C_j^{(2)}$, a higher value indicating a stronger match. For example, if $\mathcal{P}^{(2)}$ contains the same clusters in $\mathcal{P}^{(1)}$ but with permuted labels, $W$ will encode the permutation by having $w_{i,j} > 0$ if the $i$th cluster in $\mathcal{P}^{(1)}$ is the $j$th cluster in $\mathcal{P}^{(2)}$ and $w_{i,j'} = 0$ if $j' \neq j$. In general, $W$ can specify partial matching between clusters in order to handle more complicated situations, *e.g.*, $k_1 \neq k_2$, one cluster splitting into multiple clusters, etc. $W$ is solved by optimal transport (OT) with the objective of minimizing the weighted sum of distances between every pair of clusters across the two partitions. OT ensures that if the two clustering results are permuted versions of each other, the permutation will be identified through the solution for $W$.

Suppose each cluster $C_i^{(p)}$ is assigned with a significance weight $q_i^{(p)}$, with $\sum_{i=1}^{k_p} q_i^{(p)} = 1$. Usually, $q_i^{(p)}$ is the proportion of data points in $C_i^{(p)}$. Let $d(\cdot, \cdot)$ be the distance between two clusters. We solve $W$ by OT:

$$
\begin{aligned}
D\left(\mathcal{P}^{(1)}, \mathcal{P}^{(2)}\right) \triangleq \quad & \min_W \sum_{i=1}^{k_1} \sum_{j=1}^{k_2} w_{i,j} d\left(C_i^{(1)}, C_j^{(2)}\right) \\
\text{s.t.} \quad & \sum_{j=1}^{k_2} w_{i,j} = q_i^{(1)}, \forall i = 1, \ldots, k_1 \\
& \sum_{i=1}^{k_1} w_{i,j} = q_j^{(2)}, \forall j = 1, \ldots, k_2 \\
& w_{i,j} \geqslant 0, \forall i = 1, \ldots, k_1; j = 1, \ldots, k_2.
\end{aligned}
\tag{1}
$$

The Jaccard distance is adopted as the distance between clusters, i.e.,

$$d(C_i^{(1)}, C_j^{(2)}) = 1 - |C_i^{(1)} \cap C_j^{(2)}| / |C_i^{(1)} \cup C_j^{(2)}|,$$

where $|\cdot|$ means the cardinality of a set, "$\cap$" the intersection of sets, and "$\cup$" the union of sets. The first two constraints on $w_{i,j}$'s ensure that the total influence of any cluster is determined by its proportion. The objective is to minimize the weighted sum of the matching costs between clusters. The minimized objective function $D(\mathcal{P}^{(1)}, \mathcal{P}^{(2)})$ is defined as the distance between the two partitions, often called the Wasserstein distance. Consider $\mathcal{P}^{(2)}$ as the reference partition. After obtaining $W$, we normalize its $i$th row and define $\gamma_{i,j} = w_{i,j}/q_i^{(1)}$ ($q_i^{(1)}$ is the proportion of data points in $C_i^{(1)}$), which indicates the proportion of cluster $C_i^{(1)}$ mapped to cluster $C_j^{(2)}$. Denote this cluster mapping matrix as $\Gamma^{(1)} = (\gamma_{i,j}^{(1)})_{i=1,\ldots,k_1; j=1,\ldots,k_2}$.

For the general case of aligning with more than two partitions, suppose we have a reference partition $\mathcal{P}^{(r)}$ that contains $\kappa$ clusters: $C_1^{(r)}, \ldots, C_\kappa^{(r)}$. Let the proportion of points in $C_j^{(r)}$ be $q_j^{(r)}$, $j = 1, \ldots, \kappa$. Similarly, suppose we have $m$ other partitions to align with the reference, and each partition $\mathcal{P}^{(p)}$ contains $k_p$ clusters $C_1^{(p)}, \ldots, C_{k_p}^{(p)}$. Let the proportion of points in cluster $C_i^{(p)}$ be $q_i^{(p)}$, $i = 1, \ldots, k_p$, $p = 1, \ldots, m$. We align each $\mathcal{P}^{(p)}$ with $\mathcal{P}^{(r)}$. Let the cluster aligning matrix from $\mathcal{P}^{(p)}$ to $\mathcal{P}^{(r)}$ be $W^{(p)} = (w_{i,j}^{(p)})_{i=1,\ldots,k_p; j=1,\ldots,\kappa}$ and the cluster mapping matrix be

$\Gamma^{(p)} = (\gamma_{i,j}^{(p)})_{i=1,\ldots,k_p;j=1,\ldots,\kappa}$. Denote the *cluster-posterior matrix* of partition $\mathcal{P}^{(p)}$ by $P^{(p)} = (pr_{h,i}^{(p)})_{h=1,\ldots,n;i=1,\ldots,k_p}$, where $pr_{h,i}^{(p)}$ is the posterior probability of the *h*th data point belonging to cluster $C_i^{(p)}$. We then define the *aligned cluster-posterior matrix* based on $\mathcal{P}^{(p)}$ but "translated" to the cluster labels of the reference $\mathcal{P}^{(r)}$ by

$$P^{(p \to r)} = (pr_{h,j}^{(p \to r)})_{h=1,\ldots,n;j=1,\ldots,\kappa} = P^{(p)}\Gamma^{(p)}.$$

Each row of the cluster-posterior matrix $P^{(p \to r)}$ is regarded as the posterior probabilities of the corresponding point belonging to each cluster in the reference clustering result. We then use the maximum a posteriori (MAP) rule to assign the aligned cluster labels (i.e., the cluster labels used in the reference partition) to the sample points in partition $\mathcal{P}^{(p)}, p = 1, \ldots, m$. In this way, we obtain a collection of clustering results with consistent cluster labels under each view. Next, three key definitions are provided below.

**Covering Point Set (CPS).** Suppose we have already obtained the cluster mapping matrix $\Gamma^{(p)}$. Similarly, we normalize $W^{(p)}$ column-wise to obtain

$$\tilde{\Gamma}^{(p)} = (\tilde{\gamma}_{i,j}^{(p)})_{i=1,\ldots,k_1;j=1,\ldots,\kappa}, \quad \tilde{\gamma}_{i,j} = w_{i,j}/q_j^{(r)},$$

where $q_j^{(r)}$ is the proportion of data points in $C_j^{(r)}$. Based on $\Gamma^{(p)}$ and $\tilde{\Gamma}^{(p)}$, four types of topological relationships between clusters are defined: "match", "split", "merge", and "lack of correspondence". For example, $C_i^{(p)}$ and $C_j^{(r)}$ match if $\gamma_{i,j} \geqslant \zeta$ and $\tilde{\gamma}_{i,j} \geqslant \zeta$, where $\zeta$ is a relaxation threshold set between 0.5 and 1. If the "match" relationship holds between $C_i^{(p)}$ and $C_j^{(r)}$, they are considered to be the same cluster but possibly labeled differently in $\mathcal{P}^{(p)}$ and $\mathcal{P}^{(r)}$. Detailed definitions about all topological relationships are referred to [42].

Suppose for the *k*th cluster in the reference clustering result, there is a collection of matched clusters $S_i, i = 1, \ldots, m$, each is a subset of the whole dataset $\{x_1, \ldots, x_n\}$. Then the covering point set (CPS) $S_\alpha$ of cluster $k$ at a coverage level $\alpha$ is defined as the smallest set such that at least $100(1 - \alpha)\%$ of $S_i$'s are subsets of $S_\alpha$, that is, to solve the optimization problem: $min_S|S|$, s.t. $\sum_{i=1}^m 1_{(S_i \subset S)} \geqslant m(1 - \alpha)$ (we use the Least Impact First Targeted-removal algorithm developed in [42]). In summary, CPS, a counterpart of the confidence interval of a numerical estimation, is a set of possible points for one cluster at a certain level of coverage.

**Tightness.** Suppose there are *m* other partitions in total, and the proportion of partitions that have a cluster "matched" with the *k*th cluster in the reference partition is $p_k$ (*e.g.*, some partitions can have "lack of correspondence" or other relationships for reference cluster *k*). For those partitions that contain a matched cluster to cluster *k*, let the corresponding cluster *k* be sets $S_i, i = 1, \ldots, m_k, m_k \leqslant m, p_k = m_k/m$. At the coverage level $\alpha$, let $S_\alpha$ be CPS of cluster *k*. The tightness of cluster *k* is defined as

$$R_t(k) = p_k \cdot \frac{\sum_{i=1}^{m_k} |S_i|/|S_\alpha|}{m_k}.$$

Also, the overall tightness of the whole partition, denoted by $\bar{R}_t$, is defined as the average over the tightness values of individual clusters. A larger value of tightness indicates more stable clustering.

**Cluster Alignment and Points based (CAP) separability.** We first compute the CPS of each cluster in the reference partition, denoted by $S_\alpha(C_j^{(r)})$. Large overlap between the CPSs of different clusters indicates poor separation between them. The Cluster Alignment and Points

based (CAP) separability between two clusters $C_j^{(r)}$ and $C_{j'}^{(r)}$ is defined as

$$\delta_{\mathrm{cap}}\left(C_j^{(r)}, C_{j'}^{(r)}\right) = d\left(S_\alpha\left(C_j^{(r)}\right), S_\alpha\left(C_{j'}^{(r)}\right)\right),$$

where $d(\cdot, \cdot)$ can be any distance between two sets of points. We use the Jaccard distance which lies in [0, 1].

## Module 2: Form Cartesian product clusters

Suppose a collection of aligned clustering results have been obtained in each of the two views. Let the number of clusters in the first view be $\kappa_A$ and that in the second view be $\kappa_B$. Denote the reference partition for the first view by $\mathcal{A}$ and the collection of $m$ clustering results (obtained from the perturbed data) by $\mathscr{A} = \{\mathcal{A}^{(1)}, \ldots, \mathcal{A}^{(m)}\}$, where $\mathcal{A}^{(p)}$ is the $p$th clustering result and $p = 1, \ldots, m$. For brevity of notation, we record each clustering result by the cluster labels for the $n$ sample points: $\mathcal{A}^{(p)} = (a_1^{(p)}, \ldots, a_n^{(p)})$, where $a_h^{(p)} \in \{1, \ldots, \kappa_A\}$, $h = 1, \ldots, n$. Similarly, for the second view, let the reference partition be $\mathcal{B}$ and the collection of clustering results be $\mathscr{B} = \{\mathcal{B}^{(1)}, \ldots, \mathcal{B}^{(m)}\}$, where $\mathcal{B}^{(p)} = (b_1^{(p)}, \ldots, b_n^{(p)})$, $p = 1, \ldots, m$, and $b_h^{(p)} \in \{1, \ldots, \kappa_B\}$, $h = 1, \ldots, n$. After cluster alignment with the reference partition $\mathcal{A}$ (or $\mathcal{B}$), described in the previous subsection, the cluster labels in any $\mathcal{A}^{(p)}$ (or $\mathcal{B}^{(p)}$) are consistent with those used in $\mathcal{A}$ (or $\mathcal{B}$). In the subsequent discussion, we assume this is always the case.

Consider a random pair of clustering results in the two views: $\mathcal{A}^{(p_a)}$ and $\mathcal{B}^{(p_b)}$. For every point $h$, the pair of cluster labels $(a_h^{(p_a)}, b_h^{(p_b)})$ determines the Cartesian product cluster which the point belongs to. The total number of Cartesian product clusters is $\kappa_A \times \kappa_B$. We denote the Cartesian product clustering result of these two clustering results by

$$(\mathcal{A}^{(p_a)}, \mathcal{B}^{(p_b)}) = ((a_1^{(p_a)}, b_1^{(p_b)}), \ldots, (a_n^{(p_a)}, b_n^{(p_b)})).$$

We simply call $(\mathcal{A}^{(p_a)}, \mathcal{B}^{(p_b)})$ a *product partition* and $(a_k^{(p_a)}, b_k^{(p_b)})$ a *product label*. Let the Cartesian product of the two sets $\mathscr{A}$ and $\mathscr{B}$ be

$$\mathscr{A} \times \mathscr{B} = \{(\mathcal{A}^{(p_a)}, \mathcal{B}^{(p_b)}), p_a = 1, \ldots, m; p_b = 1, \ldots, m\}.$$

To reduce computation, our algorithm uses a subset of $\mathscr{A} \times \mathscr{B}$ to carry out the analysis: $\mathscr{C} = \{(\mathcal{A}^{(p)}, \mathcal{B}^{(p)}), p = 1, \ldots, m\}$. Since clustering results in different views are obtained independently, $\mathscr{C}$ is formed essentially by randomly pairing up the partitions across the two views and keeping $m$ pairs.

## Module 3: Integration across multiple views

**Optimization objective of multi-view clustering.** If we assume the clustering results in the two views are fully complementary, the product clusters induced by $(\mathcal{A}, \mathcal{B})$ (the product partition of reference clustering results in the two views) can be taken as the final clusters, an example shown in Fig 2c. In practice, however, the views are usually not fully complementary. Moreover, the number of clusters in the product partition is often too large (roughly in the exponential order of the number of views). Due to randomness in data and nuances in the clustering algorithms, an observed product cluster, *e.g.*, all the points with product label (1, 2), may not truly exist. We thus propose to merge the product clusters such that the overall tightness of the final clusters is maximized. The perturbed versions of $(\mathcal{A}, \mathcal{B})$, specifically, $(\mathcal{A}^{(p)}, \mathcal{B}^{(p)})$, provide the basis for computing the tightness of product clusters. Let the product clusters generated by $(\mathcal{A}, \mathcal{B})$ be denoted by labels $(\xi_A, \xi_B)$, $\xi_A \in \{1, \ldots, \kappa_A\}$, $\xi_B \in \{1, \ldots, \kappa_B\}$.

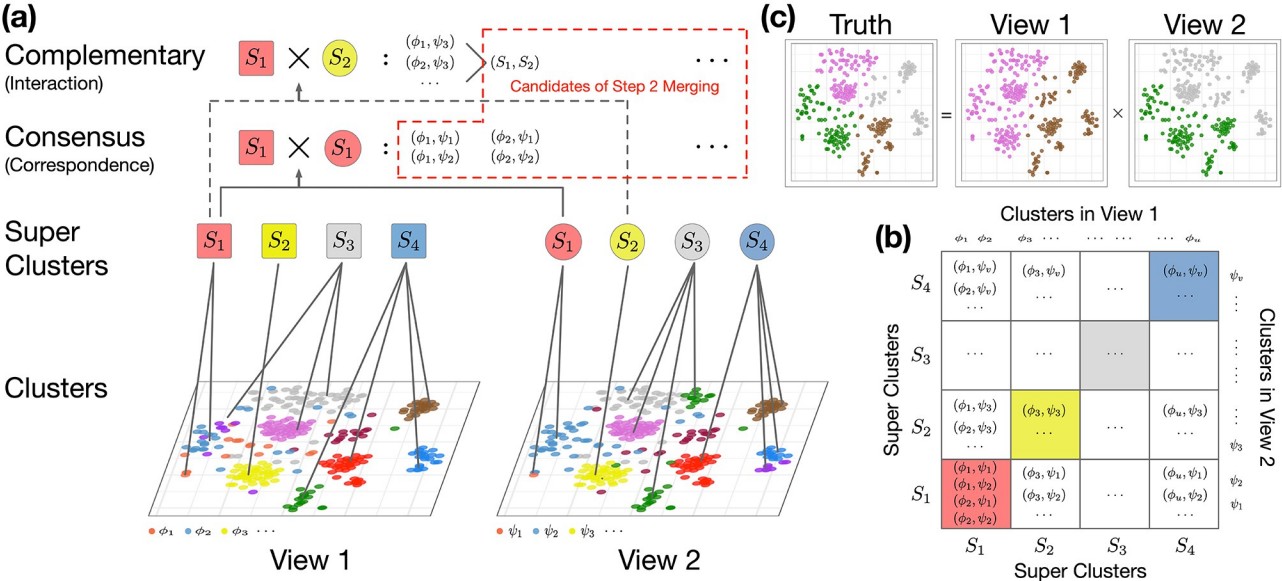

**Fig 2. First-stage merging of Cartesian product clusters based on bipartite clustering.** (a) Bipartite clustering yields super-clusters, each containing multiple clusters in every view. A super-cluster is marked by a given color, and the same super color is shown by different shapes in the two views. Any super-cluster of interaction effects will be treated as a merged product cluster in later analysis. (b) The off-diagonal white blocks correspond to unmatched product (UP) super-clusters. The diagonal colored blocks correspond to matched product (MP) super-clusters. (c) A simple case that the true clusters are the product clusters from two views. The information from the two views is fully complementary.

Suppose the desired number of clusters is $\kappa_1$. How the product clusters are merged into $\kappa_1$ clusters is given by a many-to-one mapping from a product label to a label in the set $\{1, 2, \ldots, \kappa_1\}$. Denote the mapping from the product clusters to the final clusters by $f$, where $f(\xi_A, \xi_B) \in \{1, \ldots, \kappa_1\}$. Denote the tightness of the $k$th cluster by $R_t(k)$, $k = 1, \ldots, \kappa_1$. The $k$th cluster is formed by the points in all the product clusters $(\xi_A, \xi_B)$ such that $f(\xi_A, \xi_B) = k$. Then the optimization objective is:

$$\arg \max_f \sum_{k=1}^{\kappa_1} R_t(k) \ . \tag{2}$$

The above optimization problem is intrinsically combinatorial. We thus propose a greedy algorithm that exploits a two-stage merging procedure. The first stage is optional and aims at improving computational efficiency. If the number of clusters in the product partition is small to begin with (*e.g.*, fewer than 100), we can skip the first-stage merging and thus bipartite clustering.

**First-stage: Generating and aligning super-clusters across views.** In the first-stage merging, we use bipartite clustering to generate the so-called *super-clusters*. Correspondence between clusters in different views can happen at a structural level higher than the original clusters. For instance, a cluster may split into several clusters in another view, or vice versa, multiple clusters may merge into one. Bipartite clustering aims at finding groups of clusters (aka, super-clusters) for which cross-view correspondence is sharp. With details to be explained shortly, super-clusters help decrease the number of Cartesian product clusters that proceed to the second-stage merging, thus improving computational efficiency. For large product clusters containing high proportions of data points, we determine how they aggregate mostly in the second-stage merging, while smaller product clusters are more likely to be combined based on super-clusters. Note that we do not replace the original clusters by super-

clusters. The restrictive usage of super-clusters reflects a careful balance of applying the consensus and complementary principles.

We build a bipartite graph [43, 44] for the clusters in the reference partitions $\mathcal{A}$ and $\mathcal{B}$ under the two views respectively. Let the nodes in set $\mathcal{U}$ correspond one-to-one with the clusters in $\mathcal{A}$ and the nodes in $\mathcal{V}$ correspond with those in $\mathcal{B}$. Edges exist only between a node in $\mathcal{U}$ and a node in $\mathcal{V}$. Recall that there are $\kappa_A$ clusters $\phi_1, \ldots, \phi_{\kappa_A}$ in the first view, and $\kappa_B$ clusters $\psi_1, \ldots, \psi_{\kappa_B}$ in the second view. A cluster aligning matrix $W$ (a $\kappa_A \times \kappa_B$ matrix) is computed to indicate the extent of matching between any $\phi_i, i = 1, \ldots, \kappa_A$, and $\psi_j, j = 1, \ldots, \kappa_B$. We compute $W$ using OT in the same way as that described in Section "Module 1: Generate coherent cluster labels within each view". For each $(\mathcal{A}^{(p)}, \mathcal{B}^{(p)}) \in \mathscr{C}$, let the cluster aligning matrix between $\mathcal{A}^{(p)}$ and $\mathcal{B}^{(p)}$ be $W^{(p)}$, $p = 1, \ldots, m$. We define the average of $W^{(p)}$'s, $\overline{W} = \sum_{p=1}^{m} W^{(p)}/m$, as the *matching weight matrix*. The matching weight matrix $\overline{W} = (\bar{w}_{i,j})_{i=1,\ldots,\kappa_A; j=1,\ldots,\kappa_B}$, and $\bar{w}_{i,j}$ is taken as the edge weight between nodes $\phi_i$ and $\psi_j$, a larger $\bar{w}_{i,j}$ indicating a stronger connection between $\phi_i$ and $\psi_j$. Then we use the Leiden algorithm [45, 46] for bipartite clustering. Each cluster generated in this way is what we call a super-cluster, containing in general multiple clusters in both views.

Next, we illustrate the first-stage merging with an example shown in Fig 2. Suppose 4 super-clusters $S_1, \ldots, S_4$ are formed, each containing multiple clusters in every view. For instance, suppose $\phi_1, \phi_2, \psi_1$, and $\psi_2$ belong to $S_1$ (two from each view). Product clusters formed by two cluster labels belonging to the same super-cluster—those shown in the colored diagonal blocks in Fig 2b—are kept for further analysis in the second-stage merging, *e.g.*, $(\phi_1, \psi_1)$, $(\phi_1, \psi_2)$. A product cluster will lie in an off-diagonal white block if the two cluster labels belong to different super-clusters, *e.g.*, $(\phi_1, \psi_3)$, $(\phi_2, \psi_3)$. We call $(S_i, S_j)$, $i \neq j$, an *unmatched product (UP) super-cluster*, and $(S_i, S_i)$ a *matched product (MP) super-cluster*.

In a nutshell, we will analyze the product clusters belonging to a UP super-cluster at the granularity of the super-cluster but those belonging to an MP super-cluster at the granularity of the original product clusters. Specifically, we merge all the product clusters in any UP super-cluster—they become a single cluster in the second-stage. Because of the nature of bipartite clustering, small product clusters tend to locate in UP super-clusters. For example, in Fig 2a, only the following clusters proceed to the second-stage: all $(S_i, S_j)$, $i \neq j$, and all $(\phi_m, \psi_n)$ with $\phi_m$ and $\psi_n$ belonging to the same $S_i$. Since the product clusters $(\phi_i, \psi_j)$'s capture the interaction effects between the two views, in our approach, the interaction effect between clusters in the same super-cluster will be examined at a more refined granularity.

In practice, it is possible that some product clusters are empty. Obviously, empty clusters will not feature in later analysis. Furthermore, we often observe clusters that hardly arise, which we call "rare clusters". In particular, suppose there are $m$ clustering results. If a product cluster label is not taken by sample points at least $m$ times across the $m$ results (less than one time per result on average), we say it is "rare". Points assigned with a rare cluster label are re-labeled by a majority vote. For any such point, we find its most frequent cluster label among the $m$ results and assign this label to this point in all the results.

**Second-stage: Separability-based merging to maximize tightness.** Recall that the reference partitions in the two views are $\mathcal{A}$ and $\mathcal{B}$, containing $\kappa_A$ and $\kappa_B$ clusters respectively. In addition, the two views have multiple aligned clustering results, randomly paired up to produce $m$ Cartesian product clustering results: $\mathscr{C} = \{(\mathcal{A}^{(p)}, \mathcal{B}^{(p)}), p = 1, \ldots, m\}$. For the convenience of the following discussion, suppose the first-stage merging has generated $\kappa_0$ clusters assigned with labels $1, \ldots, \kappa_0$. Let the mapping from $(\xi_A, \xi_B)$, $\xi_A \in \{1, \ldots, \kappa_A\}, \xi_B \in \{1, \ldots, \kappa_B\}$

to those $\kappa_0$ labels be $g_0(\xi_A, \xi_B) \in \{1, \ldots, \kappa_0\}$. Note that if the first-stage merging is skipped, $g_0$ is just the one-to-one identical mapping (otherwise, a many-to-one mapping). Denote the clustering result induced by $g_0$ on $(\mathcal{A}, \mathcal{B})$ by $\mathcal{C}$, which contains clusters $C_1, \ldots, C_{\kappa_0}$. We call $\mathcal{C}$ the *combined reference partition*. To generate the combined partition based on any $(\mathcal{A}^{(p)}, \mathcal{B}^{(p)})$, $p = 1, \ldots, m$, we apply OT to align the Cartesian product clusters of $(\mathcal{A}^{(p)}, \mathcal{B}^{(p)})$ with those of $\mathcal{C}$, in the same way as described in Section "Module 1: Generate coherent cluster labels within each view". Denote the $p$th combined partition obtained from $(\mathcal{A}^{(p)}, \mathcal{B}^{(p)})$ by $\mathcal{C}^{(p)}$, and let $\widetilde{\mathscr{C}} = \{\mathcal{C}^{(p)}, p = 1, \ldots, m\}$.

To solve optimization problem Eq (2), clusters in $\mathcal{C}$ are merged based on a quantity called *Cluster Alignment and Points based (CAP) separability* [42]. A higher separability corresponds to a lower similarity. These tools are collectively called *CPS (Covering Point Set) analysis*. To conduct CPS analysis, we only need cluster membership information for a collection of clustering results. In our case, the reference partition is $\mathcal{C}$ and the collection of clustering results obtained from perturbed datasets is $\widetilde{\mathscr{C}}$. Since it is computationally infeasible to examine all the possible ways of merging the clusters into $\kappa_1$ final clusters, we propose to recursively merge clusters, two at a time, in the same manner as creating a dendrogram.

**CPS merging.**   We use the pair-wise separability measure between clusters, provided by CPS analysis, as the cluster distance. We also compute the tightness of every cluster, a higher value of tightness indicating higher stability (or lower uncertainty). Suppose $C_i$ is the most unstable cluster. $C_i$ usually yields low separability from many other clusters, but the lowest pair-wise separability does not necessarily arise between $C_i$ and some other cluster. To increase the overall tightness, we first merge $C_i$ with a cluster closest to it, that is, in terms of low separability. After every merge, the per-cluster tightness and pair-wise separability measures are updated. The merging continues recursively, producing a dendrogram. In our experiment, we stop the process when the required number of clusters (usually the average tightness exceeds 0.8) is reached. The computation required is more intense than the usual way of generating dendrograms using a linkage scheme because we cannot update the separability or tightness recursively based on a linkage function. These quantities are computed from scratch after every merge. We thus have designed an accelerated version of the merging process, which is presented below.

**Accelerated CPS merging.**   At each step of merging, we set a threshold for the tightness of clusters. Any cluster with tightness below the threshold will be merged with its closest cluster. This rule essentially allows multiple merges to occur in one round without updating separability or tightness. After all such clusters are processed, we update tightness and separability measures. If some of the updated tightness measures still fall below the same threshold, we repeat the procedure, sometimes going through several rounds under a fixed threshold. If the tightness of every cluster is above the current threshold, merging can also continue if we gradually increase the threshold. In practice, the thresholds are usually set as 0.35, 0.5, 0.65, 0.8. We can apply other stopping criteria, for instance, each cluster reaching a minimum size, or a certain number of clusters having been reached (excluding singletons or tiny clusters). In most cases, the result converges at threshold 0.8 or meets another stopping criterion, *e.g.*, reaching a required total number of clusters. If computational efficiency is a concern, this accelerated merging scheme is a close substitute to the first scheme. Users also have the option to combine the two schemes, for instance, applying the second scheme first to reduce the number of clusters to a certain level and then switching to the first scheme.

After the second-stage merging, we obtain a many-to-one mapping of cluster labels from $\{1, \ldots, \kappa_0\}$ to $\{1, \ldots, \kappa_1\}$, $\kappa_1 \leq \kappa_0$, which is denoted by $g_1$. Applying $g_1$ to $\mathcal{C}$, the combined reference partition, we obtain the final clustering result, denoted by $\mathcal{F}$ that contains $\kappa_1$ clusters

$F_1, \ldots, F_{\kappa_1}$. In summary, the composite mapping $f = g_1 \circ g_0$ is a solution to the optimization problem Eq (2). Next, we present an approach to quantify the contribution of each view to the formation of any cluster. Understanding the role of each view in the generation of clusters is helpful in single-cell studies.

### Evaluating cluster-wise contribution of each view

Clusters in the final result $\mathcal{F}$ usually do not correspond well with clusters in any single view, *e.g.*, $\mathcal{A}$ or $\mathcal{B}$. We propose two methods to assess the contribution of each view to the existence of any cluster in $\mathcal{F}$. The two methods are suitable for different scenarios.

In the first scenario, we assume that final clusters $F_k, k = 1, \ldots, \kappa_1$, have been approximately captured in a single view, which generally varies with the cluster. We treat $\mathcal{F}$ as the reference partition and the raw partitions $\{\tilde{\mathcal{A}}^{(1)}, \ldots, \tilde{\mathcal{A}}^{(m)}\}$ (or $\{\tilde{\mathcal{B}}^{(1)}, \ldots, \tilde{\mathcal{B}}^{(m)}\}$) that have not been aligned with the single-view reference partition from the first view (or the second) as perturbed clustering results of $\mathcal{F}$. We can then carry out CPS analysis and compute the tightness for each cluster $F_k$. Let the tightness for $F_k$ computed from the results in the $l$th view be $R_t(k, l), l = 1, 2$. Extension to more than two views is straightforward. Then we define the contribution of the $l$th view to cluster $F_k$ by

$$\zeta_{k,l} = \frac{R_t(k, l)}{\sum_{l'} R_t(k, l')} \ .$$

If $\sum_{l'} R_t(k, l') = 0$, let $\zeta_{k,l} = 0.5$. We call $\zeta_{k,l}$ *tightness-based contribution*. The rationale for the definition of $\zeta_{k,l}$ is that if $F_k$ is stable in one view but not in another, the former view plays the dominant role in the rise of $F_k$. Apparently, the defined cluster-wise contribution of each view is between 0 and 1, a higher score indicating a higher contribution.

The definition of contribution presented above is based on the notion that the degree of uncertainty reflects the level of significance or contribution. This same concept has been utilized by existing methods that use a local weighting strategy, as seen in [47, 48]. However, these methods presuppose that different partitions are independently generated. In our scenario, since different partitions within a single view are obtained by the same method on slightly perturbed data, the independence assumption is not appropriate, making those methods unsuitable for direct application.

We note, however, $\zeta_{k,l}$ is not a good choice to quantify the contribution of each view when $R_t(k, l)$'s across all the views are low. In such a case, no cluster in any single view corresponds reasonably well with $F_k$. For instance, cluster $F_k$ is identified due to interaction effects. It is thus questionable to compare the contribution of views based on stability or tightness measures. We will use a different measure described below.

CPS analysis applied to the reference partition $\mathcal{F}$ and the partitions from the $l$th view, *e.g.*, $\{\tilde{\mathcal{A}}^{(1)}, \ldots, \tilde{\mathcal{A}}^{(m)}\}$ from the first view, provide us the cluster aligning matrix $W_l^{(p)}, p = 1, \ldots, m$, $l = 1, 2$. $W_l^{(p)}$ is a matrix of size $\kappa_{p,l} \times \kappa_1$, where $\kappa_{p,l}$ is the number of clusters in the $p$th partition from the $l$th view. Then we calculate the *cluster aligning vector* $V_l^{(p)} = \mathbf{1}_{\kappa_{p,l}}^T W_l^{(p)} / \kappa_{p,l}$ and the

*matching weight vector* $\bar{V}_l = \sum_{p=1}^m V_l^{(p)} / m$. Let the the $k$th element in $\bar{V}_l$ be $v_{k,l}, k = 1, \ldots, \kappa_1$,

which is the average matching weight of cluster $F_k$ under the $l$th view. Similar to $\zeta_{k,l}$, we define the contribution of the $l$th view to $F_k$ by

$$\eta_{k,l} = \frac{v_{k,l}}{\sum_{l'} v_{k,l'}} \ .$$

If $\sum_{l'} v_{k,l'} = 0$, let $\eta_{k,l} = 0.5$. We call $\eta_{k,l}$ *matching-weight-based contribution*. The rationale for the definition of $\eta_{k,l}$ is that if $F_k$ receives a larger weight in one view compared to another, we assume that the former view is more important for the existence of $F_k$. In our experiments, we use $\eta_{k,l}$ instead of $\zeta_{k,l}$ if more than 40% of the clusters in the reference partition have tightness 0.

## Results

In this section, we present the experimental results of CPS-merge analysis and its accelerated version (referred to as A-CPS-merge) on three multimodal scRNA-seq datasets. Table 1 summarizes basic information about the three datasets. Details on how the datasets are pre-processed are provided in their respective sub-sections. We also conducted extensive simulation studies with results provided in Table A and Table B in S1 Appendix.

We compare results with the following popular methods for multi-view clustering.

1. *Co-training clustering (Co-train)*: The EM algorithm for mixture of categorical data is used (implemented in R package `mvc` [49]).

2. *Multiple kernel clustering (MKC)*: This method is based on localized multiple kernel k-means (implemented in R package `klic` [50]).

3. *Multiple subspace clustering (MSC)*: Two-level weighted subspace clustering (implemented in R package `wskm` [51]) is used.

4. *Ensemble clustering*: This method is based on a hybrid bipartite graph formulation [52] (implemented in GitHub repository `ClusterEnsembles` [53]). We input the entire collection of clustering results used in CPS-merge analysis to the ensemble algorithm so that the comparison with CPS-merge is fair. We also performed ensemble clustering using the default input of the software and obtained similar results.

5. *Deep co-clustering (DeepCC)*: DeepCC [54] utilizes a deep autoencoder to learn a low-dimensional representation of the multi-view data, and employs a variant of Gaussian Mixture Model (GMM) for clustering (implemented in GitHub repository `Deep-Co-Clustering` [55]).

6. *Weighted-nearest neighbor (WNN)*: WNN is developed for multimodal single-cell clustering [5] (implemented in R package `Seurat` [56]). Briefly speaking, WNN generates weights for every modality based on within-modality prediction and cross-modality prediction of each cell and uses them to create a k-nearest neighbor (KNN) graph, based on which clustering is performed.

7. *Concatenation cluster analysis (CCA)*: We concatenated features from all the views. Then function `FindClusters` in the R package `Seurat` is applied to cluster the concatenated feature vectors.

We measure clustering performance by three metrics: the adjusted Rand index (ARI) [57], normalized mutual information (NMI) [58] and F-measure [59]. NMI measures the amount

**Table 1. Summary of the three multi-view datasets after pre-processing.**

| Dataset | # Instances | Dimensions (View 1, View 2) | # Clusters |
|---------|-------------|------------------------------|------------|
| HBMC | 30672 | (50, 24) | 27 |
| PBMC1 | 10032 | (50, 49) | 14 |
| PBMC2 | 161764 | (50, 50) | 31 |

of information shared by two clustering results. F-measure is the harmonic mean of precision and recall, assuming the ground truth is provided. All three metrics lie in [0, 1], with 1 indicating identical clustering. We use UMAP [60] to visualize the clustering result in each view.

The hyperparameters in our method include $\alpha$ (CPS analysis coverage level), $\delta^2$ (the variance of Gaussian noise to generate perturbed data), $\kappa_1$ (the number of final clusters), and $m$ (the number of perturbed clustering results). The effects of $\alpha$ and $\delta^2$ have been studied in [40]. We suggest $\alpha = 0.1$ and $\delta^2$ be set as 10% of the average within-cluster variance of the original data. We have also conducted a sensitivity analysis to study the performance under different values of $\kappa_1$ and $m$. Results show that the performance of CPS-merge is stable when $\kappa_1$ deviates from the true value. There is a general trend of better performance at a larger $m$ until $m$ reaches a certain level. Detailed results with discussion are provided in Table C and Fig D in S1 Appendix.

## Multimodal single-cell data

We now analyze three gold-standard multimodal single-cell datasets to demonstrate the competitive performance of CPS-merge analysis and the quantification of the cluster-wise contribution of each view. In Table 2, the performance of CPS-merge on the three datasets is compared with six other algorithms listed previously. For the competing methods, default parameter settings in the algorithms are used. The algorithm MKC becomes computationally infeasible for HBMC and PBMC2 because of the quadratic (in sample size) complexity of computing the kernel matrix. We thus cannot report its performance. For single-view clustering (View 1, View 2 in Table 2), data are pre-processed to reduce dimension before being clustered. Details on the dimension reduction methods used in each view will be provided shortly when we discuss each dataset separately. The single-view data are then clustered using the `FindClusters` function in the R package `Seurat`.

**CITE-seq Human Bone Marrow Cells (HBMC).** This dataset [61] is generated by the CITE-seq technology [62]. CITE-seq can simultaneously quantify RNA and surface protein abundance at the single-cell level by sequencing antibody-derived tags (ADTs). Thus each individual cell is measured in two views: RNA and protein (ADT). Moreover, each view individually is inadequate to identify all the cell types [61]. The data consist of 30, 672 human bone marrow cells (HBMC) of 27 different cell types.

For this example, we use one of the most popular single-cell clustering R packages `Seurat` for analyzing both views. Specifically, in each view, we follow the default Seurat clustering

**Table 2. Clustering results on three muti-view datasets (HBMC, PBMC1 and PBMC2) obtained by 8 methods (first 8 columns).** Performance is measured by ARI, NMI and F-measure. Columns View 1 and View 2 are single-view clustering results on each dataset, where View 1 refers to RNA data in datasets and View 2 refers to ADT (protein) data in HBMC and PBMC2, and ATAC data in PBMC1. The highest ARI, NMI and F-measure achieved for each dataset are in bold.

| ARI (NMI) [F-measure] | Co-train | MKC | MSC | Ensemble | DeepCC | WNN | CCA | CPS-merge | A-CPS-merge | View 1 | View 2 |
|---|---|---|---|---|---|---|---|---|---|---|---|
| HBMC | 0.695 | | 0.014 | 0.270 | 0.416 | 0.733 | 0.706 | **0.823** | 0.819 | 0.672 | 0.654 |
| | (0.774) | | (0.041) | (0.565) | (0.457) | (0.812) | (0.812) | **(0.815)** | **(0.815)** | (0.768) | (0.758) |
| | [0.723] | | [0.084] | [0.303] | [0.482] | [0.756] | [0.732] | **[0.841]** | [0.838] | [0.707] | [0.681] |
| PBMC1 | 0.635 | 0.003 | 0.241 | 0.484 | 0.203 | 0.764 | 0.744 | **0.829** | **0.829** | **0.829** | 0.668 |
| | (0.748) | (0.025) | (0.438) | (0.650) | (0.309) | (0.804) | (0.812) | **(0.839)** | **(0.839)** | **(0.839)** | (0.738) |
| | [0.673] | [0.096] | [0.320] | [0.532] | [0.289] | [0.795] | [0.776] | **[0.850]** | **[0.850]** | **[0.850]** | [0.707] |
| PBMC2 | 0.463 | | 0.006 | 0.204 | 0.142 | 0.649 | 0.620 | **0.824** | 0.764 | 0.600 | 0.643 |
| | (0.692) | | (0.011) | (0.484) | (0.310) | (0.791) | (0.761) | **(0.808)** | (0.764) | (0.740) | (0.764) |
| | [0.499] | | [0.071] | [0.241] | [0.209] | [0.678] | [0.651] | **[0.846]** | [0.796] | [0.634] | [0.675] |

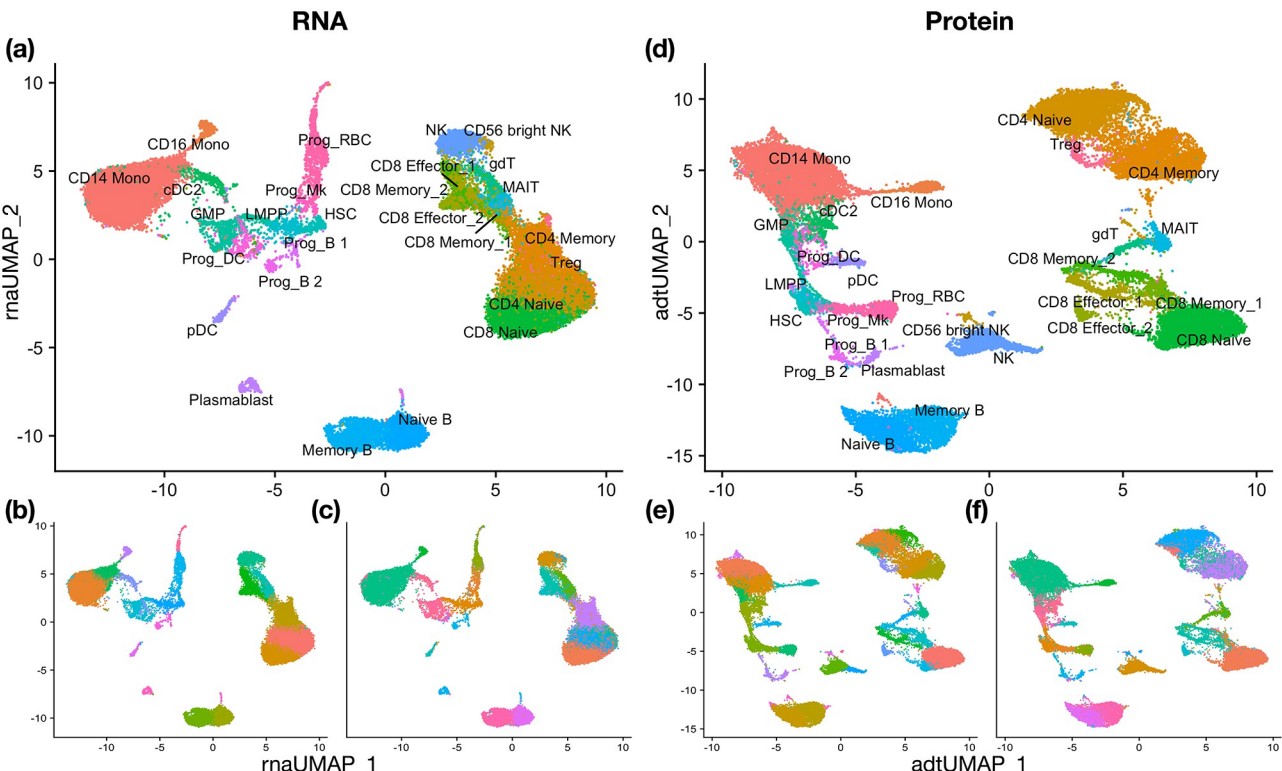

**Fig 3. UMAP visualization for HBMC data and the clustering results.** (a) True clusters on RNA. (b) Single-view clustering result on RNA. (c) CPS-merge analysis result on RNA. (d) True clusters on Protein (ADT). (e) Single-view clustering result on Protein (ADT). (f) CPS-merge analysis result on Protein (ADT).

procedure. We first log normalize the RNA data and perform the centered log-ratio transformation for ADT. We then perform dimension reduction using PCA, keeping the first 50 components for RNA and the first 24 components for ADT. For both RNA and ADT, the number of components follows the default setting in `Seurat`. Lastly, we perform cluster analysis using `Seurat` default functions `FindNeighbors` and `FindClusters`. The argument called `resolution` in `FindClusters` controls the number of clusters obtained. In multimodal single-cell data analysis, we either use the default resolution or slightly adjust it so that the number of clusters in a single view is similar to the ground-truth number of clusters. The single-view clustering results are visualized in Fig 3b and 3e. When comparing with the true cell type labels, shown in Fig 3a and 3d, we see that neither view can precisely identify all the clusters.

Results by CPS-merge analysis are shown in Fig 3c and 3f. ARI is 0.823 for CPS-merge and 0.819 for A-CPS-merge. Compared with the two single-view results, the most obvious improvement is the identification of CD14 Mono cell. As aforementioned, MKC failed to run due to the large sample size. MSC, Ensemble clustering and DeepCC yield poor accuracy. Co-train and WNN achieve relatively high ARI, but lower than that of CPS-merge analysis. CCA performs slightly better than single-view clustering, but not as accurately as CPS-merge analysis in terms of ARI.

To evaluate the cluster-wise contribution of each view, we find that more than 40% of the clusters in the final result have tightness 0, suggesting that the matching-weight-based contribution is more appropriate here. As studied in [5, 6, 63], RNA is more informative for recognizing the progenitor populations (GMP, HSC, LMPP, Prog_B1, Prog_B2, Prog_DC,

Prog_MK, Prog_RBC), while protein is more informative for distinguishing T cells (CD4 Memory, CD4 Naive, CD8 Effector_1, CD8 Effector_2, CD8 Memory_1, CD8 Memory_2, CD8 Naive, gdT, MAIT, Treg). By our analysis, the contribution of RNA to clusters corresponding to progenitor populations is on average 0.653, and the contribution of protein to clusters corresponding to T cells is on average 0.688. Therefore, our measures of the cluster-wise contribution of each view indicate that the RNA view plays a dominant role in separating progenitor populations while the protein view is more important for separating T cells. These findings are consistent with existing domain insights.

**10x Multiome Human Peripheral Blood Mononuclear Cells (PBMC1).** This dataset is generated by the 10x Genomics Multiome (https://support.10xgenomics.com/single-cell-multiome-atac-gex/datasets) ATAC + RNA kit [39], which contains 10032 peripheral blood mononuclear cells (PBMC) of 14 different cell types (Fig 4). Each cell has measurements in two views: RNA and ATAC (Assay for Transposase-Accessible Chromatin). As described in [64], ATAC-seq has much lower coverage and worse signal-to-noise than RNA-seq. Therefore, RNA provides most of the information for the clusters to be revealed, while ATAC can be used as an ancillary view. Motivated by the prior information, we use RNA as the dominant view and do not perturb the RNA data.

In each view, we follow the standard clustering procedure, which is slightly different between RNA and ATAC. For RNA data, we perform clustering as we have done with the HBMC data. For ATAC, we pre-process the data using R package `Signac` [65], which runs term frequency inverse document frequency (TF-IDF) normalization on the data and carries out dimension reduction by singular value decomposition (SVD) (we keep the 2nd to 50th

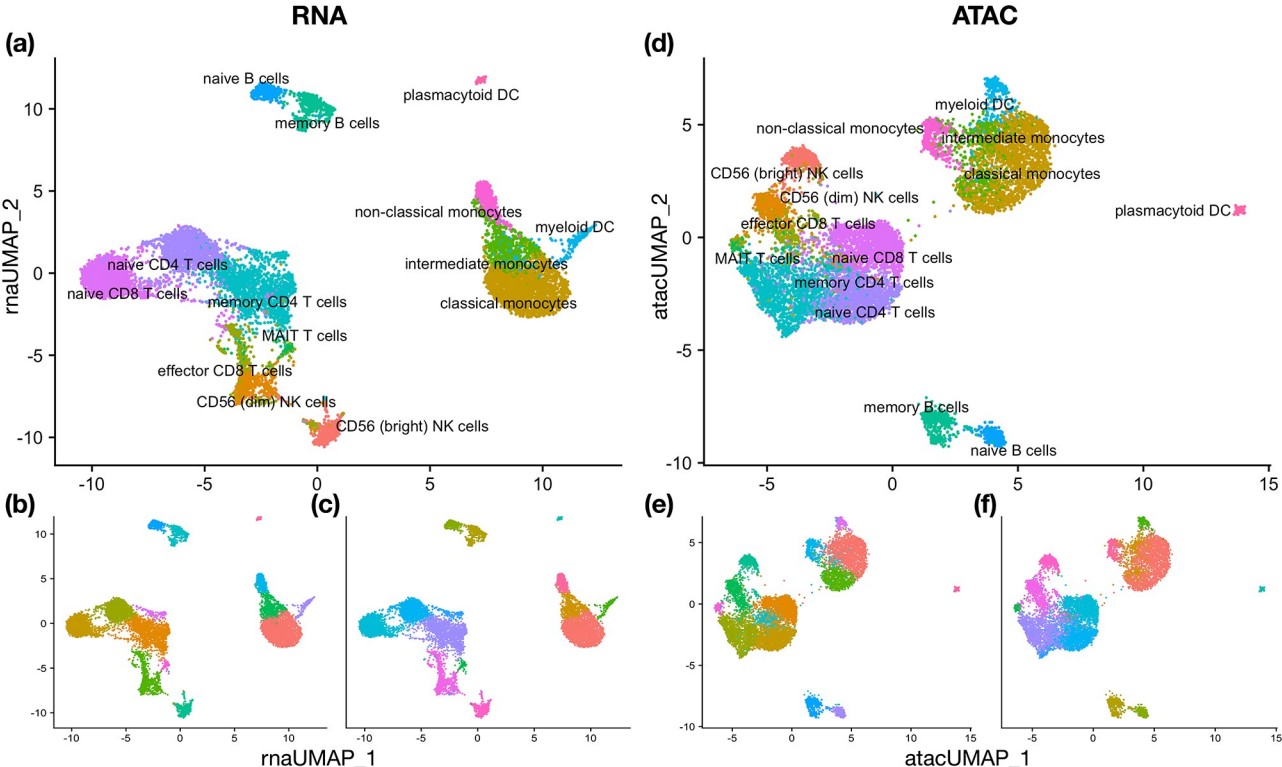

**Fig 4. UMAP visualization for PBMC1 data and the clustering results.** (a) Truth clusters on RNA. (b) Single-view clustering result on RNA. (c) CPS-merge analysis result on RNA. (d) Truth clusters on ATAC. (e) Single-view clustering result on ATAC. (f) CPS-merge analysis result on ATAC.

principal components according to `Seurat` as the first component is typically correlated with the sequencing depth). Next, we use the default `FindNeighbors` and `FindClusters` functions in `Seurat` to cluster data.

The single-view clustering results are visualized in Fig 4b and 4e. The ARI of the single-view clustering result is 0.829 for the view RNA and 0.668 for ATAC. CPS-merge and A-CPS-merge yield the same ARI of 0.829. This result suggests that ATAC do not contribute extra information about the clusters in this dataset. As shown in Table 2, Co-train, MKC, MSC, Ensemble clustering, and DeepCC all perform worse than clustering in any single view. In particular, the ARIs obtained by WNN and CCA are 0.764 and 0.744 respectively, worse than the result obtained in the RNA view. This comparison indicates that ATAC is not very useful for clarifying the true clusters.

**CITE-seq Human Peripheral Blood Mononuclear Cells (PBMC2).** The last multimodal single-cell dataset is also from peripheral blood mononuclear cells (PBMC). It is generated by CITE-seq and provided in [5]. It consists of 161, 754 cells with 31 different cell types. Same as in the previous dataset, we have two views: RNA and protein (ADT), but both views contribute substantially to the identification of clusters. This dataset has already been pre-processed. We simply apply `Seurat` to perform clustering in each view. The single-view clustering results are shown in Fig 5b and 5e. CPS-merge analysis yields an ARI value of 0.823 (A-CPS-merge analysis achieves ARI 0.764). Again, MKC failed to run because of the large sample size, and the other methods do not yield substantially better results than those obtained in any single view.

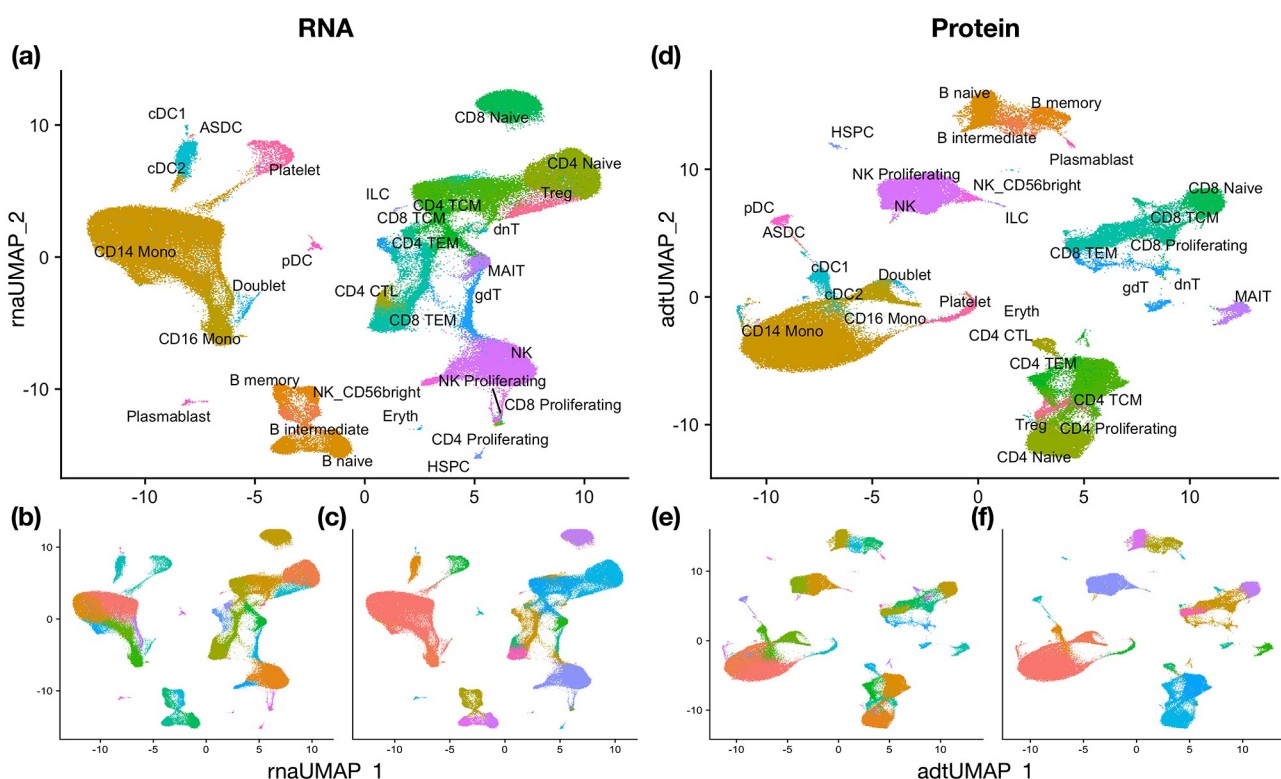

**Fig 5. UMAP visualization for PBMC2 data and the clustering results.** (a) True clusters on RNA. (b) Single-view clustering result on RNA. (c) CPS-merge analysis result on RNA. (d) True clusters on Protein (ADT). (e) Single-view clustering result on Protein (ADT). (f) CPS-merge analysis result on Protein (ADT).

As for evaluating the cluster-wise contribution of each view, we find again that more than 40% of the clusters in the final result have tightness 0. Thus the matching-weight-based contribution is more suitable to use. The contribution of the protein view to clusters corresponding to $CD8^+$ and $CD4^+$ T cells is on average 0.622, consistent with the fact that these two cell types are usually mixed in the transcriptome data but separated clearly in the protein data.

## Discussion

In this paper, we have introduced CPS-merge analysis, a new method for multi-view data clustering that is guided by both the consensus and complementary principles. As a late integration method, CPS-merge only requires cluster labels obtained from single views, making it compatible with advanced clustering algorithms designed for single-view data. We have also proposed novel measures to quantify the contribution of each view to the formation of any cluster. These measures have been validated using real datasets and domain knowledge.

However, CPS-merge has limitations in two scenarios where additional information is required for accurate results. The first scenario is when the final partition is highly unstable (*e.g.*, the average cluster tightness falls below 0.65). While such cases can be easily identified, caution is necessary when interpreting the results. The second scenario is when stable but incorrect clustering results are generated in certain single views. As our algorithm uses cluster stability to perform merging, it cannot address this issue. One potential remedy is to identify which views are ancillary a priori, allowing the algorithm to adjust accordingly.

As suggested by one reviewer, exploring online learning for multi-view clustering is a promising direction for future research. Since CPS-merge analysis only uses cluster memberships but not the original data, it can be employed in an incremental learning mode as long as the clustering algorithms used in individual views allow online learning. Numerous clustering algorithms can be easily adapted to online learning, for instance, by representing previous data using per-cluster statistics, *e.g.*, mean vectors and covariance matrices. Based on these stored representations, new data batches can be clustered or assigned to new clusters without accessing past data. Neural networks can also assist with online clustering. For instance, deep auto-encoders can encode the original data in lower dimensions, which are typically easier to cluster, particularly under an online learning paradigm. Additionally, neural networks are frequently trained in batch mode, making them naturally suited for online learning. One challenge to consider for biomedical data, such as single-cell data, is that various data batches often contain batch effects that must be eliminated. Current methods for removing batch effects typically require processing all data in one view together, preventing effective online learning. Albeit interesting, how to overcome this issue in online learning is beyond the scope of our method here.

## Supporting information

**S1 Appendix. This file contains the description of the simulation study and sensitivity analysis.**
(PDF)

## Author Contributions

**Conceptualization:** Lin Lin, Jia Li.

**Data curation:** Lixiang Zhang, Lin Lin.

**Formal analysis:** Lixiang Zhang, Lin Lin, Jia Li.

**Funding acquisition:** Lin Lin, Jia Li.

**Methodology:** Lixiang Zhang, Jia Li.

**Project administration:** Jia Li.

**Software:** Lixiang Zhang, Jia Li.

**Supervision:** Jia Li.

**Writing – original draft:** Lixiang Zhang.

**Writing – review & editing:** Lin Lin, Jia Li.

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
