## [Decision Letter · Decision Letter 0]

8 Feb 2023

Dear Professor Li,

Thank you very much for submitting your manuscript "Multi-view clustering by CPS-merge analysis with application to multimodal single-cell data" for consideration at PLOS Computational Biology. As with all papers reviewed by the journal, your manuscript was reviewed by members of the editorial board and by several independent reviewers. The reviewers appreciated the attention to an important topic. Based on the reviews, we are likely to accept this manuscript for publication, providing that you modify the manuscript according to the review recommendations.

Sincerely,

Pedro Larranaga

Guest Editor

PLOS Computational Biology

Lucy Houghton

Staff

PLOS Computational Biology

Reviewer's Responses to Questions

**Comments to the Authors:**

Reviewer #1: The topic is appropriate for publication, and the technical novelty of the paper is somewhat novel. Its contribution is moderately significant and the coverage of the problem is sufficiently comprehensive and balanced. The overall organization of the paper could be improved. The experimental results show that it obtains better performance than the state-of-the-art. However, following I have some minor questions that the authors should address to improve this work:

1. I want to advise authors to use GNN or some new methods rather than subspace learning. There are many similar methods. How to give novelty using new techniques or using the proposed methods on large-scale datasets?

2. There are many hyperparameters; how to tune them?

3. How to use the algorithm in an online learning way? Or use neural networks to optimize such ideas?

4. Somehow, it is incremental work; I still want to know the main difference and new.

5. Why the proposed method can obtain better performance than others? I would appreciate a broader discussion on why the proposed method performs better than the others.

6. I strongly recommend that authors release the source code along with the submission since the learning-based projects are typically open-source oriented to facilitate a fair assessment of the performance of the proposed methods for the community.

7. Can you give toy-data figure to show the motivation clearer?

Reviewer #2: This paper presents a late integration based multi-view clustering method for multi-modal single-cell data. In the Introduction, it should be better explained how the proposed method advances the late integration research and especially what limitations to the previous late integration based multi-view clustering methods have been tackled by the proposed method.

In page 11, the contribution of each view to the clusters in the final result is evaluated. The weighting problem has been investigated in quite a few ensemble clustering methods, such as the local weighting strategy in Locally weighted ensemble clustering and Multidiversified ensemble clustering. Please explain whether the existing weighting strategies in ensemble clustering are feasible for the proposed work and what the advantages of the proposed weighting method are.

The computational complexity of the proposed method should be analyzed. In recently, some large-scale ensemble clustering technique has been proposed, such as the ultra-scalable spectral clustering and ensemble clustering. It can be discussed whether the proposed framework can be extended to large-scale scenarios in the future work.

Some minor issues in the References: (i) For Ref. [36], it is suggested to use its journal version (https://doi.org/10.1109/TNNLS.2022.3192445) rather than its arXiv versions. (ii) For the discussions of the late integration methods in the third paragraph of the Introduction, it is strange that no references have been provided. Some late integration based multi-view clustering methods, such as [https://doi.org/10.1109/TKDE.2023.3236698], should be discussed specifically. In fact, the late integration methods merge the multiple partitions from multiple views, which resemble the ensemble clustering technique. Some discussions regarding the relationship between the late integration and the ensemble clustering are also suggested.

**Have the authors made all data and (if applicable) computational code underlying the findings in their manuscript fully available?**

Reviewer #1: Yes

Reviewer #2: Yes

PLOS authors have the option to publish the peer review history of their article (what does this mean?). If published, this will include your full peer review and any attached files.

Reviewer #1: No

Reviewer #2: No

Figure Files:

Data Requirements:

Reproducibility:

References:

---

## [Decision Letter · Decision Letter 1]

22 Mar 2023

Dear authors,

We are pleased to inform you that your manuscript 'Multi-view clustering by CPS-merge analysis with application to multimodal single-cell data' has been provisionally accepted for publication in PLOS Computational Biology.

Best regards,

Pedro Larranaga

Guest Editor

PLOS Computational Biology

Lucy Houghton

Staff

PLOS Computational Biology

Reviewer's Responses to Questions

**Comments to the Authors:**

Reviewer #1: Well addressed my concerns.

Reviewer #2: The authors have well addressed my previous concerns. I have no further comments.

**Have the authors made all data and (if applicable) computational code underlying the findings in their manuscript fully available?**

Reviewer #1: Yes

Reviewer #2: None

PLOS authors have the option to publish the peer review history of their article (what does this mean?). If published, this will include your full peer review and any attached files.

Reviewer #1: No

Reviewer #2: No

---

## [Editor Report · Acceptance letter]

4 Apr 2023

PCOMPBIOL-D-22-01585R1 

Multi-view clustering by CPS-merge analysis with application to multimodal single-cell data

Dear Dr Li,

I am pleased to inform you that your manuscript has been formally accepted for publication in PLOS Computational Biology. Your manuscript is now with our production department and you will be notified of the publication date in due course.

With kind regards,

Zsofi Zombor
